# Gut Microbial Composition Differs Extensively among Indian Native Chicken Breeds Originated in Different Geographical Locations and a Commercial Broiler Line, but Breed-Specific, as Well as Across-Breed Core Microbiomes, Are Found

**DOI:** 10.3390/microorganisms9020391

**Published:** 2021-02-14

**Authors:** Shyam Sundar Paul, Rudra Nath Chatterjee, Mantena Venkata Lakshmi Narasimha Raju, Bhukya Prakash, Savaram Venkata Rama Rao, Satya Pal Yadav, Alagarsamy Kannan

**Affiliations:** 1Poultry Nutrition Lab, ICAR—Directorate of Poultry Research, Poultry Nutrition, Hyderabad 500030, India; mvlnraju@gmail.com (M.V.L.N.R.); drbhukyaprakash@gmail.com (B.P.); svramarao1@gmail.com (S.V.R.R.); akanna72@gmail.com (A.K.); 2Director’s Lab, ICAR—Directorate of Poultry Research, Hyderabad 500030, India; rnch65@gmail.com; 3Animal Biotechnology Lab, ICAR—Directorate of Poultry Research, Hyderabad 500030, India; yadav.satyapal@gmail.com

**Keywords:** amplicon sequencing, chickens, gut microbiome, Ghagus, Nicobari, Aseel, broiler

## Abstract

Gut microbiota plays an important role in the health and performance of the host. Characterizations of gut microbiota, core microbiomes, and microbial networks in different chicken breeds are expected to provide clues for pathogen exclusion, improving performance or feed efficiency. Here, we characterized the gut microbiota of “finishing” chickens (at the end of production life) of indigenous Indian Nicobari, Ghagus, and Aseel breeds, originating from the Nicobari island, coastal India, and the Indian mainland, respectively, as well as a global commercial broiler line, VenCobb 400, using 16S rDNA amplicon sequencing. We found that diversity, as well as richness of microbiota, was higher in indigenous breeds than in the broiler line. Beta diversity analysis indicated the highest overlap between Ghagus and Nicobari breeds and a very low overlap between the broiler line and all indigenous breeds. Linear discriminant analysis effect size (LEfSe) revealed 82 breed- or line-specific phylotype operational taxonomic unit (OTU) level biomarkers. We confirm the presence of breed specific and across-breed core microbiomes. Additionally, we show the existence of breed specific complex microbial networks in all groups. This study provides the first (and comprehensive) insight into the gut microbiota of three indigenous breeds and one commercial broiler line of chickens reared without antimicrobials, and underscores the need to study microbial diversity in other indigenous breeds.

## 1. Introduction

Chickens are a cornerstone of animal agriculture worldwide, with a flock population exceeding 40 billion birds/year [1]. Poultry represents one of the most efficient form of animal protein with highly efficient feed conversion. Moreover, global human population growth, urbanization, and income levels are contributing to the huge increase in demand for protein and, therefore, livestock and poultry. Sustainable poultry meat and egg production is important to provide safe and quality protein sources in human nutrition. Feed efficiency and faster growth are crucial goals in the highly competitive poultry production system. Maintaining a healthy gut is an important prerequisite to attain these goals.

The gastrointestinal (GI) tracts of chickens are densely populated, with diverse and complex microbiota (bacteria, fungi, archaea, protozoa, and virus—dominated by bacteria) that play a vital role in the digestion and absorption of nutrients, host immune system development, pathogen exclusion, and endocrine activity; thus, maintaining normal physiological homeostasis and influencing gut development, nutrient supply, and host metabolism and detoxification [2]. Understanding diversity and community structure of the gut microbiome is important for devising strategies to improve chicken gut microbiome. Several studies have indicated that the gut microbial composition in chickens is strongly influenced by the host [3,4,5,6]. Further, host immune pathways in gut tissues may influence microbiome structure. It has been shown that chicken lines selected for differences in immune response contribute to adaptive changes in gut microbiota [7]. Similarly, some studies have indicated a strong correlation between the gut microbiome and feed efficiency in chickens [8,9]. In studies with cattle, it has been shown that a heritable set of core gut microbiome influence dairy cow productivity [10]. It has been suggested that high-throughput sequencing tools have huge potential to be used for assessing the microbiome structure in the guts of chickens, in a comprehensive manner, which, in turn, may further develop strategies to improve growth, feed efficiency, survivability, or lower pathogen shedding via the development of host specific probiotics [11].

Native chicken breeds are gaining popularity across India due to their unique characteristics, such as desirable flavor of meat and eggs and ability to thrive in a low input system and hot humid climate. Further, there is a growing demand in cities for eggs and meat of native chickens, as compared to the exotic ones, due to the perceived better flavor and taste. Many producers are rearing indigenous chicken breeds under intensive systems to meet increasing demand.

Some of the indigenous chicken breeds of India including Aseel, Nicobari, and Ghagus, have been shown to have better egg, meat traits, and resistance to infectious diseases, with better immune parameters, but lower feed conversion efficiency, as compared to commercial broiler chickens [12]. Analysis to define gut microbiome of native Indian breeds, such as Assamese chicken [13], Aseel, and Kadaknath has recently been reported [6]. However, the study on Aseel and Kadaknath was carried out when chickens were supplemented with antibiotic growth promoters. Antibiotics are known to modulate gut microbiota significantly and, hence, information on the true composition of gut microbiota without influence of antibiotics of these breeds remains to be studied.

In the present study, we use amplicon sequencing targeting hypervariable (V3-V4) region of 16S rRNA genes to compare gut microbiota of finishing chickens (at end of production life) of three indigenous Indian breeds originating in diverse geographical regions (Aseel, Ghagus, and Nicobari) and a commercial broiler (Vencobb 400) reared under commercial set-up, but without any antibiotics. The Nicobari breed is an indigenous and endemic breed of chickens of the Nicobar Islands, an internationally acknowledged biodiverse hot spot off the Indian mainland (it was detached from the Asian main lands some 100 million years ago). The breed produces the highest number of eggs among all indigenous chicken breeds of India [14] and is believed to have the high disease-resistance [15]. Aseel is an important and one of the most popular indigenous chicken breeds of the Indian mainland, well known for its meat quality, with desirable taste and flavor, as well as the ability to thrive under adverse, climatic, and nutritional conditions [16]. Ghagus is another important native chicken breed of the coastal region of Southern India, known for its meat quality, with desirable taste and flavor, disease resistance [15], ability to perform under hot and humid climates, and under the low plane of nutrition.

Our hypothesis is that gut microflora would vary significantly between chicken breeds and lines, offering clues for development of breed/line specific probiotics or feed additives for improving performance, feed efficiency, health or pathogen exclusion—or for targeted genetic improvement—by selective breeding for the desirable type of gut microbiome.

## 2. Materials and Methods

### 2.1. Chicken Breeds and Experimental Design

In the present study, four chicken breeds or lines were chosen for comparison, which included three indigenous Indian breeds (Nicobari, Aseel, and Ghagus) and one global commercial broiler line (Vencobb 400). The chickens of the indigenous breeds utilized in the present study were reared at the farm of Indian Council of Agricultural Research (ICAR)—Directorate of Poultry Research (DPR) under intensive system. These breeds were originally collected from their respective breeding tracts (Nicobari from Andaman and Nicobar Island, Ghagus from Kolar district of Karnataka State, and Aseel from Andhra Pradesh) approximately six years ago. The birds of native breeds utilized in this study were hatched in the institute (DPR) hatchery (300 birds/breed) through random breeding of 60 sires and 240 dams from each breed. The first 16 weeks after hatch, the chickens were housed breedwise, separately, on conventional floor pens (600 ft^2^), on fresh and clean paddy husks. There was free access for chickens to antibiotic and coccidiostat free feed and water. Brooding was done with the help of incandescent bulb, up to 21 days. From 17 to 35 weeks, each chicken was reared in an individual cage (2 ft^2^) with arrangements for individual feeder and nipple drinkers. All chickens of native breeds were allowed ad libitum feeding maize and soya bean-based balanced diets, as per institute (ICAR-DPR) developed feeding standards for intensive production with no antimicrobials. All of the native chickens were offered a chick diet (179.4 g/kg crude protein (CP) and 2880 Kcal metabolizable energy (ME)/kg) from hatch to 8 weeks of age, a grower diet (177.7 g/kg CP and 2810 Kcal ME/kg) from 9 to 16 Weeks of age, followed by a breeder diet (140 g/kg CP and 2870 Kcal ME/kg). Detailed compositions of the diets are presented in Appendix A. All birds were vaccinated against viral diseases, such as Marek’s disease, Newcastle disease, infectious bursal disease, and fowl pox, as per the recommended schedule.

VenCobb400 line (Venkateswara Hatchery and Breeding Farm Private Limited, Hyderabad) was used for comparison as representatives of the global commercial broiler line. Newly hatched Vencobb 400 chicks were acquired from a local hatchery (Venkateswara Hatchery, Hyderabad, India) and reared on clean battery brooder cages with five birds/pen measuring 6 ft^2^, with the arrangement for feeder and drinker. Brooding was done with the help of incandescent bulbs up to 21 days. All of the chickens of the Vencobb 400 line received maize and soya bean meal-based balanced diets, as per feeding standards. Chickens had free access to antibiotic- and coccidiostat-free feed and water. The chickens were fed with a pre-starter diet (229.3 g/kg CP and 2950 kcal/kg ME) from hatch to 14 d, a starter diet (214 g/kg CP and 3100 Kcal/kg ME) from 15 to 28 d, followed by a finisher diet (195 g/kg CP and 3250 kcal/kg ME from 29 d onward, as per feeding standards recommended by the breeder. Detailed compositions of the diets are presented in Appendix A. All birds were vaccinated against viral diseases, such as Marek’s disease, Newcastle disease, and infectious bursal disease, as per the recommended schedules.

Care was taken to ensure that all birds received, exactly, similar husbandry to minimize non-host variation. Birds of each breed or line were housed in separate distantly placed houses so that there was no fecal contamination between pens.

### 2.2. Sample Collection and DNA Extractions

For each breed/line, eight apparently healthy male chickens in their finishing (marketable age) stage, with body weight close to mean body weight (BW) (average BW, 2.4 kg; age, indigenous: 35 weeks; broiler, 6 weeks) were selected at random (each one from separate pens in case of Vencobb 400) from each group, caught, and euthanized by cervical dislocation. Gut was opened immediately using sterile scissors; luminal contents of hindgut (from duodenum to cloaca including caeca) were recovered into sterile cryovials, mixed thoroughly, and immediately stored in a portable freezer at −20 °C, transported to the laboratory and stored at −80 °C.

Total genomic DNA was extracted from the pooled gut contents of each individual chicken using the commercially available QIAamp Fast DNA Stool Mini kit (QIAGEN, Hilden, Germany), following the manufacturer’s Instructions. DNA concentration and quality were assessed using a Qubit 2.0 fluorometer (Invitrogen, Thermo Fisher Scientific, Waltham, MA, USA) and gel electrophoresis. DNA was stored at −20 °C until further processing.

### 2.3. 16S rRNA Gene Amplification and Sequencing

Hypervariable V3 and V4 regions within the 16S rRNA gene were amplified from gut microbial DNA sample using the primer pair S-D-Bact-0342-b-S-17(5′-CCTACGGGNGGCWGCAG-3′) and S-D-Bact-0785-1-A-21(5′-GACTACHVGGGTATCTAATCC-3′) recommended by Klindworth et al. [17] with adapters (for forward primer: 5′-TCGTCGGCAGCGTCAGATGTGTATAAGAGACAG-3′; for reverse primer: 5′-GTCTCGTGGGCTCGGAGATGTGTATAAGAGACAG-3′). Each 25 μL PCR reaction comprised of 2.5 μL DNA (~5 ng/μL), 5 μL each forward and reverse primer (1 pM) and 12.5 μL 2X KAPA HiFi HotStart ReadyMix (Kapa Biosystems, London, UK). PCR amplification cycles were as follows: initial denaturation at 94 °C for 3 min, followed by 25 cycles of 95 °C for 30 s, 55 °C for 30 s, and 72 °C for 30 s, and a final extension at 72 °C for 5 min. Amplicons were further processed for library preparation using Illumina’s Nextera XT library preparation kit (Illumina, San Diego, CA, USA). Sequencing was performed using an Illumina MiSeq desktop sequencer. Trimming of adaptor sequences was performed using Illumina analysis software V2.5 as recommended by the manufacturer using default parameters. Samples were processed with three negative controls per plate in the sequencing run.

### 2.4. Sequence Data Analysis

Illumina reads were analyzed using MOTHUR software package (v 1.40.0) [18] by following analysis pipeline of MiSeq SOP (https://www.mothur.org/wiki/MiSeq_SOP; accessed on 31 March, 2020) with some modifications. Briefly, read pairs were assembled into contigs. A threshold of Phred quality scores (Q ≥ 25) of the base was chosen for a stringent quality control processing. Any contigs with ambiguous base (N), having homopolymer bases greater than 8 and shorter than 200 bp were culled. Identical or duplicate sequences were merged. Sequences were aligned to SILVA [19] seed alignment (silva.seed_v138.align; available in the MOTHUR website). Poorly aligned sequences were removed and overhangs at both ends were trimmed so that they overlap the same region. Unique sequences were screened and further de-noised based on pre-clustered command for up to 2 bp differences between sequences. Chimera sequences were checked and removed using VSEARCH [20], as implemented in MOTHUR. Sequences were then classified using naïve Bayesian classifier against ribosomal database project (RDP) 16S rRNA gene training set (version 16) with bootstrap cutoff of 51% [21]. Sequences classified to unrelated taxon (other than bacteria or Archaea) were removed. Clean sequences were subjected to operational taxonomic unit (OTU) clustering using DMSC software [22] at 97% similarity cutoff. DMSC output was converted to MOTHUR formatted list file and shared file for further analysis. OTUs were taxonomically classified using RDP classifier as implemented in MOTHUR using the GreenGene [23] Taxonomy (gg_13_8_99.gg.tax) database, available in the MOTHUR website. MOTHUR formatted shared file and consensus taxonomy files were converted to BIOM file format, and singleton and doubleton OTUs were removed from the BIOM file. The BIOM file along with the sample metadata files were uploaded to the MetaCoMET web server [24] for plotting Venn diagram of OTUs. The breed specific OTU tables along with metadata were uploaded to the METAGENassist website [25] for analysis, where data were filtered as per default settings, and samples were normalized using total sum (sample vs. sample) and Pareto scaling (taxon vs. taxon; mean centered and divided by the square root of standard deviation of each variable). The processed data were used for the generation of correlation heatmaps using the Spearman rank option. The BIOM (or MOTHUR generated shared and consensus taxonomy files) along with the metadata file and a neighbor joining (NJ) tree file prepared from the OTU representatives were uploaded to MicrobiomeAnalyst [26] for the analysis of alpha diversity, beta diversity, differential abundance, biomarker identification, core microbiome, and correlation network analysis. For the analysis of alpha diversity and beta diversity (nonmetric multidimensional scaling (NMDS) and principal coordinate analysis (PCoA)), data were normalized by the cumulative sum scaling (CSS) method after disabling the default data filtering options for low counts and low variances. However, features appearing only in one sample were removed automatically as there was no option to disable such minimal filtering function. Additionally, for the analysis of alpha diversity, data were rarefied to the minimum library size (at 12151 sequences per sample). For the analysis of differential abundance (using edgeR), biomarker identification (using linear discriminant analysis effect size or LEfSe) and correlation network data were filtered for low counts and low variances using the default setting to remove less informative features, to focus on important features, and to improve downstream statistical analysis (this process removed 127 low count OTUs out of 296 taxonomic OTUs). Beta diversity profiling and significance testing were carried out at different taxonomic levels, such as OTU, genus, family, class, and phylum using PCoA as well as NMDS ordination based on different distance methods, such as Bray–Curtis dissimilarities, Jensen–Shannon diversion, and Weighted UniFrac, using statistical methods, such as permutational multivariate analysis of variance (PERMANOVA) and homogeneity of group dispersion (PERMDISP). Core microbiome analysis was carried out at OTU level using MicrobiomeAnalyst with relative abundance cutoff of 0.1% [6,27], and a prevalence cutoff of 50% applied, instead of default values of 0.01% and 20%, respectively, to focus on the most commonly shared OTUs. On detection of significant difference in overall abundance between groups on edgeR analysis, followed by Benjamini–Hochberg false discovery rate (FDR) correction for multiple comparison, groups were compared pairwise using nonparametric Mann–Whitney U test (Wilcoxon rank sum test) as implemented in SPSS [28]. Chicken breed/line-specific biomarkers at multiple taxonomical level analyses were performed using the linear discriminant analysis (LDA) effect size (LEfSe) algorithm using Benjamini–Hochberg false discovery rate (FDR) adjusted *p*-value cutoff value of 0.05 and the logarithmic LDA score cutoff of 2, as well as 3.5 [29]. The LEfSe bar plots were created using MicrobiomeAnalyst, but the cladogram was created using a standalone version of LEfSe [29]. In all analyses, *p*-values were corrected for Benjamini–Hochberg false discovery rate (FDR). BIOM data were rarefied to the minimum library size (at 12151 sequences per sample) and rarefaction analyses were carried out using MOTHUR. Rarefaction curve was visualized by creating plots using R package. Alpha diversity matrices were compared at the OTU level using the Kruskal–Wallis test followed by Dunn’s test, as implemented in SPSS [28], and Benjamini–Hochberg false discovery rate (FDR) adjustment of *p*-values. Correlation networks at different taxonomic levels were built based on the pairwise Spearman rank correlation coefficients, where each node represents a taxon, and two taxa are connected by an edge if the Spearman rank correlation between the two taxa meet the *p*-value (<0.05) and correlation (>0.6) thresholds. Taxonomic assignments were presented as Krona charts from CSS normalized relative abundance data using standalone KronaTools (version 2.7.1) [30]. Stacked barplots of taxonomic assignments at the phylum level were created using MicrobiomeAnalyst.

## 3. Results

### 3.1. Microbiome Sequencing

High throughput sequencing generated 5.095 million raw reads corresponding to 4.08 Gbp of raw data from the gut content of the 32 chickens. After read quality filtering, merging paired-end reads, denoising, removing chimeras, and filtering low-quality sequences, the average number of quality-controlled sequences per sample was 113,028 (range, 50,392–295,955) (Table 1). The 16s rRNA gene amplicon sequencing results were deposited in the Sequence Read Archive of the National Centre for Biotechnology Information (NCBI) (accession numbers: PRJNA641245 and PRJNA641779).

### 3.2. Operational Taxonomic Unit (OTU) Occurrence

The indigenous Aseel breed presented the highest number of non-singleton non-doubleton (with >2 members) OTUs (genetic distance based OTUs at 97% similarity cutoff) (3461), followed by Ghagus (1911) and Nicobari (1767) (Table 1). The commercial broiler line presented the lowest number of OTUs (857). Observed OTU numbers were significantly (*p* < 0.05) higher in the indigenous breeds than that of the commercial broiler line. Observed OTU numbers in the Aseel breed were higher than those of Ghagus and Nicobari. However, OTU numbers in Ghagus and Nicobari were comparable.

Based on Good’s coverage index, 98.6, 99.1, 97.2, and 97.7% of gut microbial diversity were covered in Broiler, Aseel, Ghagus, and Nicobari, respectively (Table 1).

### 3.3. Taxonomy Assignment

The assignment of consensus taxonomy resulted in the identification of 18 phyla, 138 genera, and 296 phylotype-OTUs (phylotype-OTUs were obtained after merging distance-based OTUs with the same consensus taxonomy) being represented across the gut samples of the chicken population. Appendix A (Krona charts and stacked barplots) provides an overview of the average CSS normalized relative abundance levels of most of the abundant microbiota at different taxonomic levels in the gut microbiota of different breeds/lines. In Aseel, the gut microbiota was dominated by Bacteroidetes (44%) followed by Firmicutes (43%), Proteobacteria (6%), Actinobacteria (1%), and Cyanobacteria (0.8%) that constituted 94.8% of the whole phyla. In the broiler line Firmicutes, Bacteroidetes, Cyanobacteria, Proteobacteria, and Verrucomicrobia were the major phyla, which accounted for 81, 6, 6, 4, and 1% of total sequences, respectively. In Ghagus Bacteroidetes, Firmicutes, Proteobacteria, and Cyanobacteria were the major phyla, which accounted for 62, 26, 3, and 0.9% of total sequences, respectively. In Nicobari Bacteroidetes (53%), Firmicutes (24%), Proteobacteria (8%), Fusobacteria (5%), Verrucomicrobia (2%), and Cyanobacteria (2%) were predominant phyla. Overall, the two phyla, Firmicutes and Bacteroides, represented 77–88% of gut prokaryotes.

A Venn diagram depicting the extent of overlap of phylotype-OTUs between different hosts has been presented in Appendix A. Aseel, Broiler, Ghagus, and Nicobari had 263, 172, 249, and 230 phylotype-OTUs, respectively. Only 27 phylotype-OTUs were specific to any breed or line (Aseel: 25; Broiler: 1; Ghagus: 8; Nicobari: 3), whereas, 35, 89, and 135 phylotype-OTUs were shared between 2, 3, and 4 groups, respectively.

### 3.4. Microbial Alpha Diversity and Data Rarefaction

Different alpha diversity metrics (the diversity within each breed or line) were estimated based on rarified data to assess different aspects of the community structure and the results are presented in Figure 1 and Table 2. The mean observed richness (number of observed OTUs) was comparable among native breeds, but was lower in the broiler line. The species richness (or the number of species or OTUs) indices, such as abundance based coverage estimator (ACE) and Chao1, were higher (*p* < 0.01) in the indigenous breeds than that of the commercial broiler line. Indigenous breeds had comparable ACE and Chao1 estimates.

The diversity (which takes into account both richness and evenness) estimators, such as Simpson, Shannon, and Fisher were also higher (*p* < 0.01) in the indigenous breeds (Aseel, Ghagus, and Nicobari) as compared to the commercial broiler line, and these estimators were comparable among the indigenous breeds.

Besides richness and diversity estimators, rarefaction curves based on the Chao1 index were also plotted. The rarefaction curve depicts the correlation between the number of sequences and the number of OTUs and the steeper the slope, the higher the diversity [31]. The rarefaction curve also indicated that the broiler line had lower diversity than those of the indigenous breeds (Figure 2). Rarefaction curve approached the asymptotic level for each breed or line, suggesting the availability of sufficient reads to represent each microbial community.

### 3.5. Microbial Beta Diversity

The beta diversity (the partitioning of biological diversity among breeds or along a gradient, e.g., the number of species shared between two breeds or lines) analysis was undertaken to assess the relationship of microbial communities of different breeds/line using different metrics to calculate the dissimilarity/distance matrix, such as Bray–Curtis, Jensen–Shannon, unweighted UniFrac, and weighted UniFrac.

The correlation between the distance matrix and metadata categories was tested using PERMANOVA, which reports an coefficient of determination or R squared (*R*^2^ ) value indicating the proportion of variation explained by this category, and a *p*-value representing the statistical significance [32]. Homogeneity of group dispersions was also tested using PERMDISP. Beta diversity was visualized using NMDS as well as PCoA but due to space limitation only plots obtained using NMDS are presented. Results of beta diversity analysis at phylotype-OTU level, including results of ordination using NMDS, are presented in Figure 3.

PERMANOVA tests performed using all beta diversity metrics used in this study showed significant (*p* < 0.001) differences in community structure between different breeds/lines, both at Phylotype-OTU level and at the Phylum level (Appendix A). At phylotype-OTU level, Jensen, Shannon based PERMANOVA analysis had the highest pseudo–F (11.56) and *R*^2^ (0.553) values among all four distance metrics indicating that 55.3% of microbiota variation is explained by this category (breed) besides a significant *p*-value (*p* < 0.001). The weighted UniFrac based analysis at the phylotype-OTU level showed that breed explained 47.3% (*R*^2^) of microbial variation (PERMANOVA, pseudo–F 8.4, *p* < 0.001). At the phylum level, Jensen Shannon based PERMANOVA analysis had the highest Pseudo- F (18.54; *p* < 0.001) and *R*^2^ (0.665) value among all four distance metrics, indicating that 66.5% of microbiota variation is explained by this category (breed) besides a significant *p*-value (*p* < 0.001). The weighted UniFrac based analysis at the phylum level showed that breed explained 52.7% (*R*^2^) of microbial variation (PERMANOVA, pseudo–F 10.4, *p* < 0.001). The beta dispersion values (PERMDISP) were non-significant for all groups in all diversity metrics analyzed at phylotype-OTU or the phylum level, except in the case of unweighted UniFrac analysis for phylum data, indicating homogeneous dispersion among groups.

Beta diversity plots visualized using ordination methods NMDS at the phylum level using NMDS method of ordination have been presented in Appendix A. Jaccard index resulted in similar plots in NMDS scaling as that of Bray–Curtis distance both at phylotype-out, as well as at Phylum Level and, hence, plots for the Jaccard index have not been presented.

The NMDS scaling based on all five distance metrics showed clear visual separation of breed/line at phylotype-OTU level. At the phylum level, there was a high degree of overlap between the indigenous breeds, but the only minor overlap between Broiler and indigenous breeds (Aseel or Nicobari or Ghagus) was observed, and the extent of overlap between breeds/line varied with the distance metric used.

### 3.6. Differential Abundances at Different Taxonomic Levels

The 157 phylotype-OTUs (with ≥4 members and prevalence in >20% samples) were taxonomically placed (using RDP classifier and Greengenes database) into a total of 91 genera with 88, 69, 88, and 87 genera in Aseel, broiler, Ghagus, and Nicobari, respectively. At the family level, sequences were classified into a total of 68 families with 66, 54, 65, and 68 families in Aseel, broiler, Ghagus, and Nicobari, respectively. At the order level, sequences were classified into a total of 44 orders with 42, 32, 42, and 44 orders in Aseel, broiler, Ghagus, and Nicobari, respectively. At class level, sequences were classified into a total of 35 classes with 34, 29, 35, and 35 classes in Aseel, broiler, Ghagus, and Nicobari, respectively. At the phylum level, sequences were classified into a total of 20 phyla with 19, 18, 20, and 20 phyla in Aseel, broiler, Ghagus, and Nicobari, respectively.

Out of 157 phylotype-OTUs, edgeR analysis with FDR correction indicated that 88 phylotype-OTUs were significantly different in abundance between breeds/line. Major phylotype-OTUs (top 41 out of 88 phylotype-OTUs in term of CSS normalized abundance) with significant difference in abundances between breeds/line along with taxonomy (last classified level) has been presented in Figure 4. Many phylotype-OTUs containing a large number of sequences had low taxonomic resolution (having a taxonomic resolution only down to the order level). Comparisons between pairs of groups using the Mann-Whitney U test indicated that among the 41 most abundant phylotype-OTUs there was a significant difference in abundance of 24, 19, 21, 24, 26, and 6 phylotype-OTUs between Aseel vs. broiler, Aseel vs. Ghagus, Aseel vs. Nicobari, broiler vs. Ghagus, respectively. Interestingly, the abundance of many phylotype-OTUs such as OTU1000062 (order Bacteroidales), OTU100296 (family equivalent *S24-7* uncultured gut microbial group), OTU102407 (genus Bacteroides), OTU1057116 (phylum Bacteroidetes), OTU168571 (species Bacteroides barnesiae), OTU1758401 (genus equivalent SMB53 uncultured gut group), and OTU4324240 (genus Faecalibacterium) and were very low in the broiler line although these were highly abundant in all indigenous breeds. On the other hand, abundance of OTU137026 (genus Lactobacillus), OTU1021172 (species *Lactobacillus salivarius*), OTU137026 (species *Lactobacillus agilis*), OTU181074 (genus equivalent *CC115* gut group), and OTU549991 (species *Lactobacillus helveticus*) were higher in the broiler line than those of indigenous breeds.

Many phylotype-OTUs were significantly higher in abundance in Aseel than in Ghagus or Nicobari. The notable ones, besides others, include OTU1000113 (order Burkholderiales), OTU100567 (genus Ruminococcus), OTU839684 (family Lachnospiraceae), OTU586453 (family Christensenellaceae), OTU1010876 (genus Oscillospira), OTU1057116 (phylum *Bacteroidetes*) OTU 167741 (genus Dorea), and OTU1649772 (species *Escherichia coli*). On the other hand, abundance of OTU1021172 (species *Lactobacillus salivarius*) and OTU1066621 (genus Prevotella) were higher in Ghagus than in Aseel. The abundance of a few phylotype-OTUs, such as OTU100296 (family *S24-7*) and OTU1758401 (family *SMB53*) were lower in Nicobari than in Aseel or Ghagus. The abundance of OTU4369050 (family Fusobacteriaceae) was higher in Nicobari than in Aseel or Ghagus. The abundance of OTU1066621 (genus Prevotella) was significantly higher in Ghagus than in Aseel or Nicobari.

Out of 91 genera, having a mean abundance of ≥4 and prevalence of >20%, twenty-four genera were significantly different in relative abundance between breeds/line. Genera having significant differences in abundance between breeds/lines have been presented in Appendix A. Out of 45 orders, meeting the minimum count and prevalence criteria, twenty-two orders were significantly different in relative abundance between breeds/line. Orders having a significant difference in abundance between breeds/lines have been presented in Appendix A. Phylum level abundance data of gut microbiota have been presented in Appendix A. Out of 19 Phyla, edgeR analysis followed by FDR correction indicated that abundances of five phyla were significantly different between groups. Comparisons between different pairs of groups using the Mann–Whitney U test indicated a significant difference in abundance of 3, 0, 5, 3, 3, and 1 phylum between Aseel vs. broiler, Aseel vs. Ghagus, Aseel vs. Nicobari, broiler vs. Ghagus, broiler vs. Nicobari, and Ghagus vs. Nicobari groups, respectively. The broiler line had a significantly higher abundance of Firmicutes as compared to indigenous breeds. Indigenous breeds had a significantly higher abundance of unclassified Bacteroidetes as compared to the broiler line. Some of the phyla, such as Deferribacteres, Elusimicrobia, Spirochetes, *TM7,* and unclassified *WPS2* were either not detected or detected in a few samples in the broiler but were detected consistently although in low numbers in indigenous breeds. At the phylum level, there was a significantly higher abundance of Actinobacteria, unclassified Bacteroidetes, Firmicutes, and TM7 in Aseel as compared to Nicobari. There was no difference in the abundance of any phylum between Excel and Ghagus. The abundance of phylum TM7 was significantly higher in Ghagus than in Nicobari.

### 3.7. The Core Gut Microbiome

Phylotype-OTUs with a mean normalized (CSS) relative abundance of at least 0.1% and having within breed prevalence of at least 50% in at least one breed were considered to define core microbiome in different breeds/lines. Analysis of the prokaryotic community composition at the OTU level indicated that of the 296 phylotype-OTUs detected, only 22 phylotype-OTUs were present in >0.1% relative abundance in at least 50% of the birds in at least one breed or line (Figure 5). Twelve, five, and two of the 22 phylotype-OTUs belonged to the phyla Firmicutes, Bacteroidetes, and Proteobacteria, respectively, with the rest belonging to other diverse phyla, such as Fusobacteria and Cyanobacteria. Only five phylotype-OTUs (OTU1000113 belonging to the order Clostridiales, OTU100567 belonging to the genus Ruminococcus, OTU1010876 belonging to the genus Oscillospira, OTU839684 belonging to the family Lachnospiraceae, and OTU98948 belonging to the family Ruminococcaceae) occurred as core microbiome across all breeds or lines and these core phylotype-OTUs represented 26% of total microbial count. The OTU839684 belonging to the family Lachnospiraceae alone accounted for 13% of the total microbiome (or 50% of core microbiome) count of all chicken breeds/lines. Three phylotype-OTUs were unique to the broiler line (OTU181074 belonging to the genus level group *cc_115*, OTU2229500 belonging to the species *Subdoligranulum variable*, OTU549991 belonging to the species *Lactobacillus helveticus*). Two phylotype-OTUs were unique to the Aseel breed (OTU586453 assigned to the family Christensenellaceae and OTU60774 assigned to the species *Anaerobiospirillum thomasii*). One phylotype-OTU was unique to the Nicobari breed (OTU4369050 belonging to the genus Fusobacterium). Seven phylotype-OTUs (OTU1000062 belonging to the order Bacteroidales, OTU100296 belonging to the family level gut group *S24_7*, OTU102407 belonging to the genus Bacteroides, OTU1057116 belonging to the phylum Bacteroidetes, OTU1105376 belonging to the genus Sutterella, OTU168571 belonging to the species *Bacteroides barnesiae,* and OTU4324240 belonging to the species *Faecalibacterium prausnitzii*) were core OTUs across the three indigenous breeds, but were not consistently detected in the broiler line. These indigenous chicken specific core OTUs represented 49.3% of the total prokaryote count of indigenous birds.

### 3.8. The Breed/Line Specific Biomarkers Based on LEfSe Algorithm

The LEfSe analysis identified biomarkers in the gut microbiota (a specific tax that varies in abundance consistently by chicken breed or line) that were indicative of gut microbiota of each breed or line. In total, 82, 35, 54, 37, 27, 21, and 14 biomarkers were identified with LDA scores >2 at phylotype-OTU, species, genus, family, order, class, and phylum levels, respectively. A high abundance of genera Bacteroides, Oscillospira, Faecalibacterium, Coprococcus, Anaerobiospirillum, Sutterella, Olsenella, Paraprevotella, unclassified Clostridiaceae, Cloacibacillus, Turicibacter, Treponema, Collinsella, Succinatimonas, Gemmiger, Methanobrevibacter, and Desulfovibrio were typical for Aseel breed (Figure 6a).

A high abundance of genera or genus equivalent taxonomic groups Lactobacillus, Ruminococcus, unclassified Lactobacillaceae, Subdoligranulum, Dorea, cc_115, Blautia, Escherichia, Clostridium, Bilophila, Defluviitalea, Bifidobacterium, and Eggerthella were typical for the broiler line. Similarly, a higher abundance of few genera or genus equivalent taxonomic groups namely SMB53, Prevotella, Odoribacter, Parabacteroides, Akkermansia, YRC22, and RFN20 were typical to the Ghagus breed. Higher abundance of Fusobacterium, Megamonas, Asteroleplasma, Barnesiella, Helicobacter, Elusimicrobia, WCHB1_41, Desulfovibrio, and Spirochaetes were typical to the Nicobari breed.

A cladogram of important biomarkers identified at different taxonomic levels in different breeds/lines using LefSe with LDA scores >3.5 has been presented in Figure 6b. The class *Coriobacteriia*, orders, such as unclassified Bacteroidetes and Aeromonadales, families, such as Christensenellaceae, Ruminococcaceae, and Succinovibrionaceae, were major biomarkers in the Aseel. The phylum Firmicutes, class Bacilli, order Enterobacteriales, and families, such as Lactobacillaceae and Enterobacteriaceae, were top biomarkers in the broiler line. The phylum Bacteroidetes, classes TM7_3, Bacteroidia, and Lentisphaeria, orders, such as Bacteroidales, and families, such as Odoribacteraceae, Paraprevotellaceae, Porphyromonadaceae, Prevotellaceae, S24_7, Clostridiaceae, and Victivallaceae were top biomarkers in Ghagus. The phylum Elusimicrobia and Synergistetes, classes Fusobacteria, Elusimicrobia, Synergistia, Verruco_5, Epsilonproteobacteria, Betaproteobacteria and Pedosphaerae, orders, such as Elusimicrobiales, Fusobacteriales, Burkholderiales, and families, such as Bacteroidaceae, Elusimicrobiaceae, Fusobacteriaceae, Alcaligenaceae, and Synergistaceae were top biomarkers in Nicobari.

### 3.9. Correlation Analysis

Family level correlations among microbes in Aseel, broiler, Ghagus, and Nicobari are shown in Appendix A. For Aseel, the occurrence of families that include potentially pathogenic species, such as Enterobacteriaceae, Clostridiaceae, Campylobacteraceae, Pasteurellaceae, Streptococcaceae, Staphylococcaceae, Fusobacteriaceae, Enterococcaceae, Corynebacteriaceae, and Helicobacteraceae exhibited a positive correlation with each other, and a high negative correlation with Christensenellaceae and Ruminococcaceae and a low correlation with Lactobacillaceae and Bifidobacteriaceae (Appendix A).

In the case of the broiler line, several small family level clusters with a strong positive correlation with each other were detected. Campylobacteraceae, Fusobacteriaceae, Prevotellaceae, Rikenellaceae, Acidaminococcaceae, Bacteroidaceae, and Desulfovibrionaceae exhibited a strong positive correlation with each other and a negative correlation with most other families. Enterobacteriaceae, Pseudomonadaceae, and Coriobacteriaceae showed a strong positive correlation with each other (Appendix A).

In Ghagus, also, several family level clusters of prokaryotes having strong positive correlations among themselves were detected, such as Lactobacillaceae, Desulfovibrionaceae, and Enterobacteriaceae; Succinovibrionaceae, Helicobacteraceae, Fusobacteriaceae, Bifidobacteriaceae, and Peptostreptococcaceae, etc. (Appendix A).

In Nicobari, the families, such as Methanocorpusculaceae, Victivallaceae, Fusobacteriaceae, Campylobacteraceae, Prevotellaceae, Veillonellaceae, Verrucomicrobiaceae, and Bacteroidaceae exhibited a strong positive correlation with each other with a negative correlation with Coriobacteriaceae, Enterobacteriaceae, Erysipelotrichaceae, Lachnospiraceae, Leuconostocaceae, Clostridiaceae, Hyphomicrobiaceae, and Lactobacillaceae (Appendix A).

## 4. Discussion

The advent of high throughput sequencing and omics approaches as tools for the study of microbial communities has allowed a detailed characterization of the gut microbiota of chickens in a quick and robust fashion, without the need to culture the microorganisms. This is the first study exploring gut microbiome of Nicobari and Ghagus breeds of Chicken breeds originating in Nicobari islands (a well-known biodiversity hotspot) and coastal India, respectively. Here, we attempted to identify differences in the gut microbial community structure of native chicken breeds originating in diverse geographical locations and with a global commercial broiler line at the end of their productive life (finishing stage).

The microbial communities differ through the chicken’s gastro intestinal tract with particular microbial profiles detected in crop, gizzard, ileum, cecum, and colon of broiler chickens [33]. Here we analyzed microbiota from the entire hindgut (duodenum to cloaca including caecum) to focus on segments generally considered to be most important for gut health and function. Within hindgut segments, population density in caecal content is generally higher than other segments. Samples from entire hindgut was mixed and, hence, it is likely that cecal microbiota might have dominated over other segments. However, considering that we had modest sequencing depth, it can be safely assumed that microbes from all gut segments are adequately represented. Further, as we have mixed the gut content similarly for all of the groups, the mixing is unlikely to impact results with respect to ability to compare community composition and diversity between breeds or lines.

The initial colonization of the gastrointestinal tract of birds occurs naturally after hatching and can even begin before, by passing of microorganisms through the pores of the eggshell [34]. After the initial colonization of the intestine, the species richness and complexity of the population structure of the microbiota increases as the birds grow, until microbiota reaches a state of stabilization. Based on results from multiple studies [35,36,37,38] Diaz Carrasco et al. [39], while reviewing the subject noted that this process normally occurs in commercial broiler chickens around 3 weeks of life. However, they also noted that development times and succession patterns of intestinal microbiota species could vary, depending on the genetic makeup of the birds and management factors. Here, we selected time points late in the production cycle of each breed or line (as each breed matures, i.e., at marketing age) to permit assessment of the outcome of colonization throughout each chicken’s productive life. In the present study, there was considerable difference in age between the broiler and native chickens (5 weeks vs. 35 weeks). The rate of growth and rate of passage of digesta are faster in the case of broiler as compared to native birds. It is difficult to find an ideal equivalent age to compare fast growing broilers with slow growing native chickens. Studies on broiler gut microbiota beyond 6 weeks age are scarce and may not be of much interest to stakeholders as these birds are unlikely to be reared beyond 5–6 weeks. On the other hand, at 5 or 6 weeks, native birds would have much lower BW and probably immature gut than broilers of same age. Our intent was to compare the microbiome profile of these two groups of birds at marketable age (at the end of their productive life), as this is essential to help design strategies to modulate composition—not only to improve host health and performance—but also to control zoonotic agents that can contaminate poultry products (thus, posing a risk to consumer health) and, in turn, reduce use of antibiotics. Earlier, Ocejo et al. [40] sampled chickens of fast growing broilers (Ross 308) and a slow growing free-range chicken breed (Sasso-T451A) at different ages (42 d and 86 d, respectively) to define and compare their microbial community structure at the end of their productive life.

In the present study, the indigenous chicken breed presented a higher number of OTUs (1767 to 3461) than that of the commercial broiler line (857). In contrast, in an earlier study the commercial broiler line VenCobb 400 was reported to have more OTUs (1273) than Indian indigenous chicken breeds, such as Aseel (735 to 1134) or Kadaknath (816 to 833). In general, more OTUs were detected in this study as compared to that of Pandit et al. [6]. This may be partly attributed to the fact that, in the current study, no antibiotic growth promoter (AGP) was used, and we analyzed the gut content from the entire hindgut (duodenum to cloaca including caeca), whereas in the study reported by Pandit et al. [6], only cecal content of bird-fed diets containing AGP was analyzed.

A total of 135 (out of 296) phylotype-OTUs (with >2 members) were shared by chickens from all groups.

The gut microbiota of Indian native chicken breeds evaluated here were dominated by sequences belonging to the phyla Bacteroidetes and Firmicutes whereas the gut microbiota of the commercial line was dominated by sequences representative of the phylum Firmicutes. Within Indian indigenous breeds, Firmicutes/Bacteroidetes ratio were more and less similar in the Ghagus and Nicobari breeds, but substantially different from that of Aseel. Our data are in discordance with a previous report by Saxena et al. [13], showing the dominance of Firmicutes in gut microbiome of the Assamese breed of Indian Indigenous chickens. Pandit et al. [6] reported that Bacteroidetes was the dominant phyla in most of the gut caecal samples of Indian indigenous breeds, such as Kadaknath, and Aseel and Firmicutes were more common in Cobb 400 samples, which is in concordance with our current study. Generally, higher Firmicutes/Bacteroidetes ratios are shown to correlate with obesity in humans [41]. Both Firmicutes and Bacteroidetes are primarily carbohydrate fermenters. Firmicutes are known to produce both butyrate and propionate, whereas Bacteroidetes primarily produce propionates, such as fermentation end product [42]. Within Firmicutes, different genera under the order Clostridiales were predominant in different breeds or line. The Bacilli members, like *Lactobacillus* spp., possessing prebiotic and probiotic activities, were present in very high proportions in the broiler line but were in small proportions in indigenous breeds. However, the genus Faecalibacterium (under order Clostridiales and family Ruminococcaceae), known to produce butyrate and, thus, have a crucial role in maintaining gut health and host well-being [43], were present in significant proportion in indigenous breeds, but not in the broiler line. The genus Bacteroides and an unknown genus under the order Bacteroidales constituted the top two dominant genera in all of the indigenous breeds, whereas Lactobacillus and an unknown genus under the family Lachnospiraceae constituted the top two dominant genera in the boiler line. It has been shown that Faecalibacterium co-occurs with several members of Bacteroidetes in gut [44] and it has been suggested that Faecalibacterium may rely on Bacteroides for cross-feeding [43]. Interestingly, as in case of broiler, an unknown genus under the family Lachnospiraceae constituted a very high proportion (11%) of microbiota in Aseel, but were detected in relatively lower proportion in other indigenous breeds. Gut microbial composition is mediated by many factors, such as geographical location, host diet, and administration of antibiotics and other medicines. It has been shown that the succession of changes in gut microbiota correlates with changes in the cytokine profile expressed by host intestinal cells [35].

In the present study, alpha diversity estimators measuring species richness (Chao1and ACE) as well as diversity (Simpson, Shannon, and Fisher) and rarefaction curve indicated that the broiler line VenCobb 400 had lower diversity than those of the indigenous breeds, in disagreement with a previous study [6], where both slow growing native breeds and a broiler line were sampled at the same (42 d) age. Our study is consistent with two previous studies [40,45], which presented higher richness and complexity of the gut microbiome in slow growing free range chicken breeds than in the broiler when sampled at their respective finishing age, or at random, irrespective of age. Since age is considered as one of the factors that are known to influence gut microbiota composition [40], part of the observed difference between the broiler line and native breeds may also be attributed to the difference in age. Nevertheless, the present study shows that species richness and complexity in hindgut microbiota of three Indian native breeds (Aseel, Ghagus, and Nicobari) are significantly higher than that of the broiler chicken line Vencobb 400 at the finishing stage of their life.

Beta diversity analysis involving different distance metrics has indicated clear separation of microbiome at both OTU as well as phylum levels by chicken breed or line, indicating a strong host component in microbiome composition, in agreement with previous studies [6,46,47]. Low levels of overlap between microbiome of the broiler line with those of indigenous breeds also suggest a highly different microbial composition across these two categories. Interestingly, NMDS plots also indicate that compositional distribution of gut microbiome of the island breed (Nicobari) have very high degree of similarity with both the coastal breed (Ghagus) and the breed from Indian mainland (Aseel), indicating possibility of strong evolutionary linkage among all of these indigenous breeds.

Differential abundance analysis using edgeR indicated that abundances of several phylotype-OTUs belonging to the order Bacteroidales, and a phylotype-OTU under genus Faecalibacterium were significantly lower in the broiler line VenCobb 400. However, abundances of few phylotype-OTUs belonging to the genera, such as Lactobacillus, Bilophila, Clostridium, Weissella, Eggerthella, and Defluviitalea were higher (*p* < 0.05) in the broiler line as compared to indigenous breeds.

At order level 18, 6, 12, 14, 17, and 1, orders were differentially abundant between Aseel vs. broiler, Aseel vs. Ghagus, Aseel vs. Nicobari, broiler vs. Ghagus, broiler vs. Nicobari, and Ghagus vs. Nicobari, respectively. A similar trend was also observed at other taxonomic levels, indicating highest overlap in microbial community composition between the coastal breed (Ghagus) and the island breed (Nicobari) among indigenous breeds and a very low overlap between the broiler line and all indigenous breeds.

LEfSe biomarkers also indicated a possible strong host genetic influence on gut microbiome.

Breed- or line-specific potential pathogenic and/or zoonotic organisms were also detected as biomarkers, such as Bilophila, Escherichia, Clostridium and Eggerthella in broiler; Anaerobiospirillum, Sutterella, Turicibacter, Collinsella, and Treponema in Aseel; SMB53 gut microbial group in Ghagus; Fusobacterium, Asteroplasma, Helicobacter, and Elusimicrobia in Nicobari.

On the other hand, many breed- or line- specific prokaryotes having potential beneficial effects on gut health of hosts (probiotic or butyric acid or short chain fatty acid producing properties) were also associated as biomarkers, such as Faecalibacterium [48], Coprococcus [49], Gemmiger [50] in Aseel; Lactobacillus [48], Subdoligranulum [48], Dorea [51], Blautia [48], and Bifidobacterium [52] in the broiler line; Akkermansia [53] in Ghagus; Barnesiella [54] in Nicobari.

Using 0.1% relative abundance and 50% prevalence cutoff criteria, 22 phylotype-OTUs out of 296 phylotype-OTUs representing 26% of the total microbial count were found to qualify as core OTUs in at least one breed or line. In contrast, only five phylotype-OTUs representing 13% of total microbiome occurred as core microbiome across all breeds or line.

Interestingly, few potentially pathogenic or harmful (Sutterella in Aseel and Nicobari, Fusobacterium in Nicobari, and Anaerobiospirillum in Aseel) prokaryotes were detected above the abundance and prevalence cutoffs; thus, qualifying to be part of the core microbiome in different breeds or lines. This is in line with an earlier report where Campylobacter was detected above the 1% cutoff in Kadaknath and Aseel, but not in Cobb400 or Ross 308 [6].

In contrast, many prokaryotes having potential beneficial effects on gut health of the hosts (having probiotic or butyric acid producing properties) were also detected above the 0.1% relative abundance and 50% prevalence cutoff in different breeds and lines.

Further work will be required to ascertain the exact role of the pathogenic bacteria detected as biomarkers or core microbiome. However, besides being opportunistic or primary pathogens, inflammatory and immune responses induced by these pathogens have been suggested to influence the intestinal environment, host immunity, and its bacterial communities [55,56].

Despite inter-individual differences in community composition, a core set of microbes shared across individuals of a particular breed, suggesting presence of breedwise distinct community composition. Millions of years of co-evolution between the host and microbes have led to a mutualistic symbiosis in which the microbiota contributes to many host physiological processes, and the host, in turn, provides a nutritious and hospitable environment to the microbes. Further, the normal gut symbionts forms a stable community that resists the invasion and colonization of non-native bacteria [57].

Potentially opportunistic pathogens, such as those within the families Enterobacteriaceae, Clostridiaceae, Campylobacteraceae, Fusobacteriaceae, etc., exhibited a strong correlation with each other and a negative correlation with beneficial bacteria, having probiotic (competitive exclusion) properties or short chain fatty acids (SCFA) production ability, belonging to families, such as Christensenellaceae, Lactobacillaceae, and Bifidobacteriaceae. Several breed- or line-specific microbial family level clusters with strong positive correlations with each other and negative correlations with other clusters were detected in all of the breeds or lines, suggesting the existence of strong interactions among different microbial groups. Correlation network analysis data, at genus and order levels, further substantiates existence of diverse and complex microbial networks. Similar results were reported in few earlier reports [6,58].

The present study indicates existence of breed specific gut microbial structure, which will be useful in the development of breed- or line-specific feed additives. In the future, further efforts should be made to characterize gut microbiome of other major breeds or lines of chicken.

## 5. Conclusions

In this study, we carried out the first comprehensive analysis of the chicken gut microbiome of the Nicobari and Ghagus breeds of Indian native chickens originating in biodiverse hotspots, such as Nicobari Island and coastal India, respectively. We carried out a comparative analysis of diversity and composition of gut microbiota of three indigenous breeds and one commercial broiler line. This study provides important insights into chicken breed- or line-specific variations in enteric bacterial occurrences, diversity, and complex microbial networks. Our amplicon sequencing results emphasizes more similarity of the microbiota within the gut lumen of indigenous breeds as compared to the commercial broiler line VenCobb 400, but exhibits distinctive taxonomic differences between them as well. The study also indicates that, among native breeds, there is more similarity of the gut microbiome of the island breed (Nicobari) with that of the coastal breed (Ghagus), as compared to the mainland breed (Aseel). This study indicates the existence of breed- or line-specific core microbial as well as across-breed or line core microbiome in chickens, and the occurrence of beneficial and potentially opportunistic pathogenic microbes as part of the core microbiome. A deeper understanding of host–microbiome interactions, as emanated from the current study, is expected to support the development of strategies, including the development of breed-specific feed additives and probiotics for enhanced productivity from unconventional or low-value diets, to prevent colonization by pathogenic and zoonotic organisms, and to develop an alternative to antibiotic growth promoters.

## Figures and Tables

**Figure 1 microorganisms-09-00391-f001:**
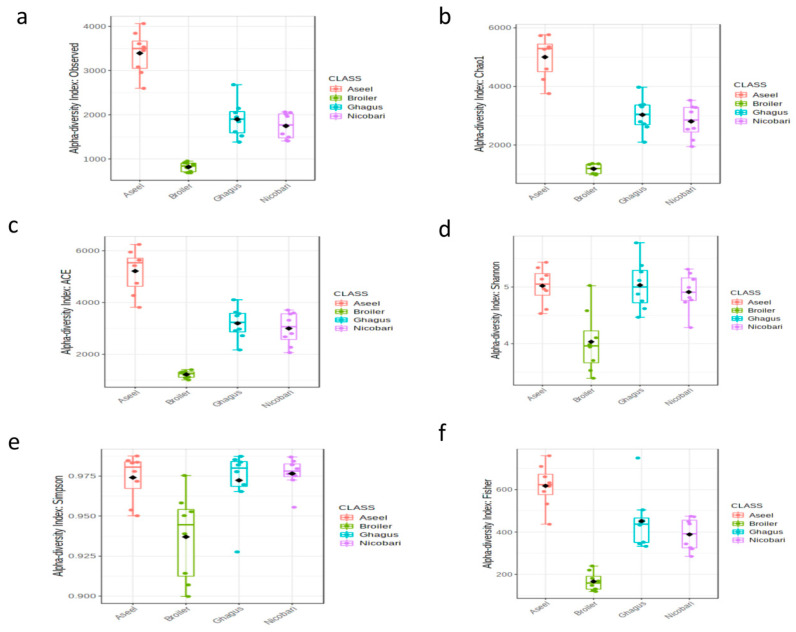
Alpha diversity across samples. Sequences were rarified to the minimum library size (at 12,151 sequences per sample). Differences in alpha diversity metrics of microbial diversity and richness in the gut microbiota of chickens of different breeds or lines are shown as boxplots. (**a**) the observed number of OTUs, (**b**) Chao1, (**c**) abundance based coverage estimator (ACE), (**d**) Shannon, (**e**), Simpson, (**f**) Fisher.

**Figure 2 microorganisms-09-00391-f002:**
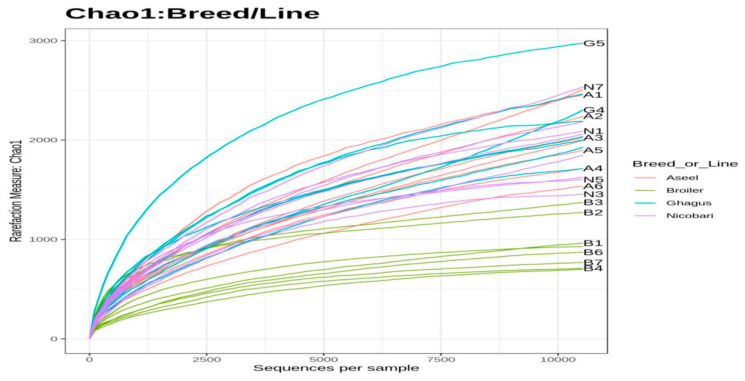
Rarefaction curves based on Chao1. Sequences were rarefied to the minimum library size (at 12,151 sequences per sample).

**Figure 3 microorganisms-09-00391-f003:**
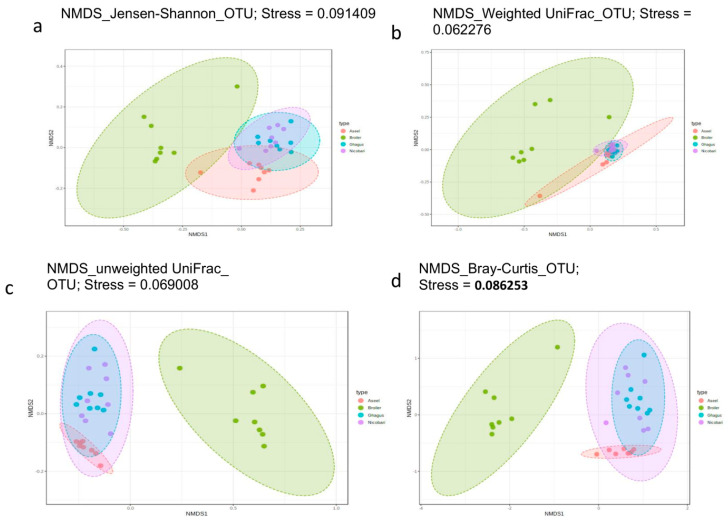
Beta diversity among breeds or lines. Beta diversity plots visualized using nonmetric multidimensional scaling based ordination at OTU level for different beta diversity metrics (**a**) Jensen-Shannon, (**b**) weighted UniFrac, (**c**) Un-weighted UniFrac, (**d**) Bray-Curtis. A stress value of less than 0.1 represents a high-quality ordination.

**Figure 4 microorganisms-09-00391-f004:**
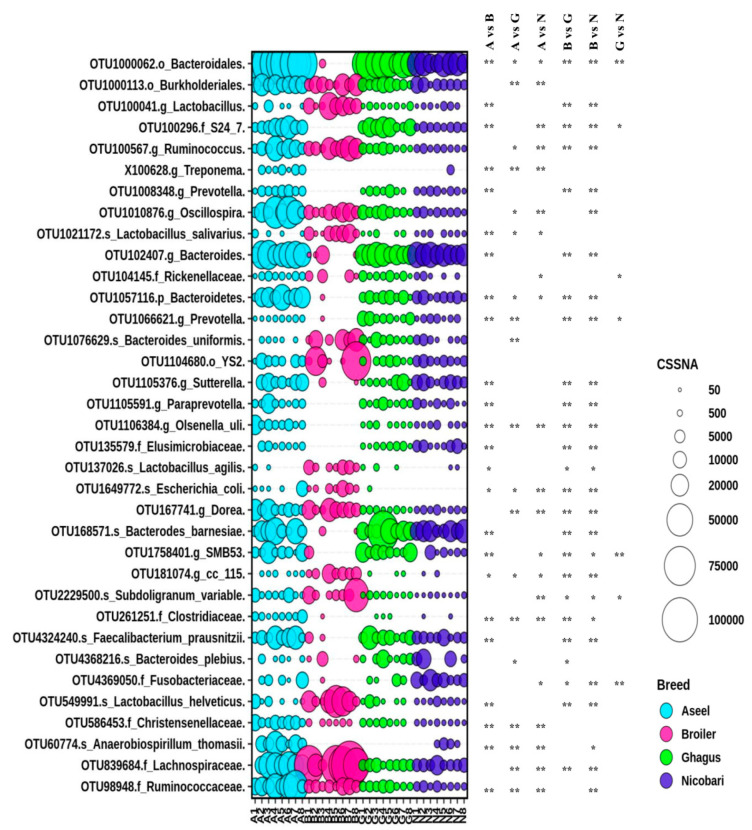
Differential abundance of gut microbiota in different breeds or lines at OTU level. Top 35 abundant phylotype-OTUs, out of 88 phylotype-OTUs with a significant difference in relative abundance among groups identified with edgeR, were plotted. The size of bubbles in the bubble plot indicates the normalized (cumulative sum scaling) abundance of each OTU. * *p*-value < 0.05; ** *p*-value < 0.01.

**Figure 5 microorganisms-09-00391-f005:**
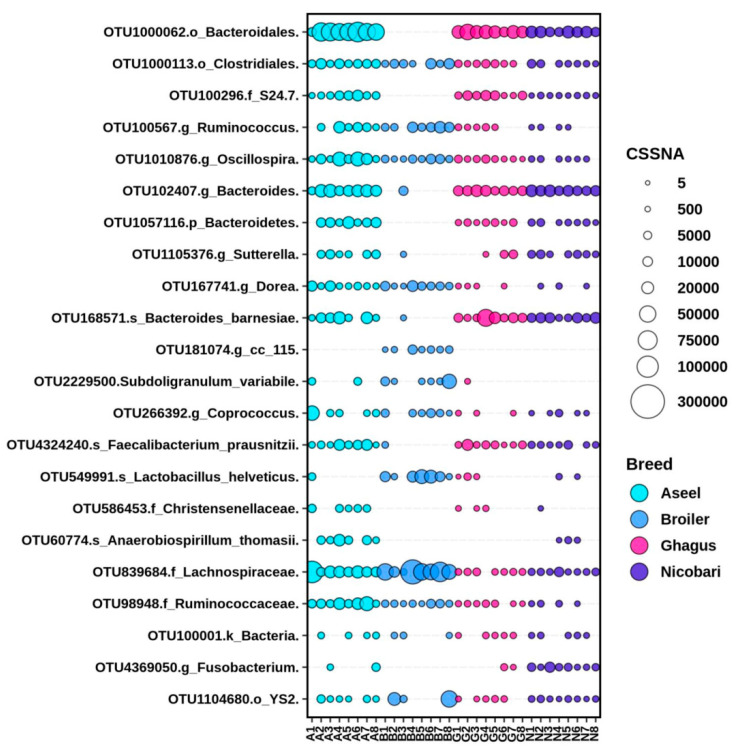
Breed or line-specific and across-breed or line core phylotype-OTUs. Phylotype-OTUs with a mean normalized (cumulative sum scaling) relative abundance of at least 0.1% and having within breed or prevalence of at least 50% in at least one breed were taken into account The size of bubbles in the bubble plot indicates normalized (cumulative sum scaling) abundance of each OTU.

**Figure 6 microorganisms-09-00391-f006:**
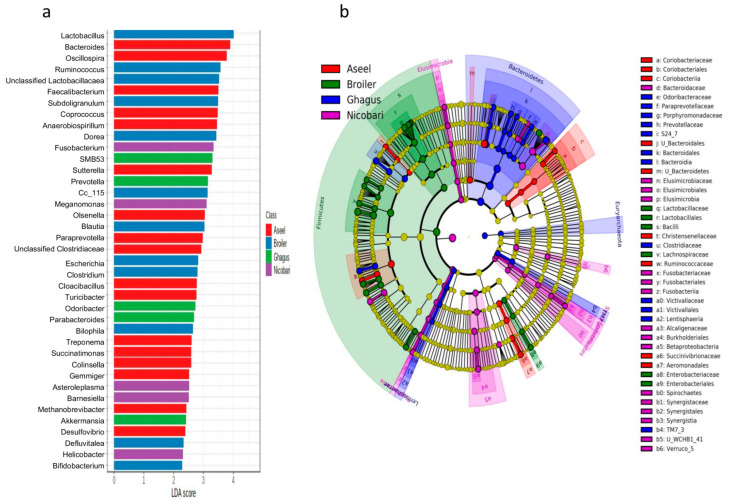
Chicken breed or line-specific biomarkers. (**a**) Genus level biomarkers identified using linear discriminant analysis effect size (LEfSe) analysis using the Kruskal–Wallis test (*p* < 0.05) with linear discriminant analysis (LDA) score >2.0. (**b**) Cladogram representation of differentially abundant microbiota at the different taxonomic levels. The taxonomic levels of the phylum are labeled, while the order to the genus is abbreviated (only labels of top 43 clades are shown here), with the colors indicating breed/line with the highest abundance. The cladogram has been dual rooted to denote domain archaea and bacteria.

**Table 1 microorganisms-09-00391-t001:** Summary statistics of sequences analyzed, operational taxonomic unit (OTU) numbers detected (clustered at >97% similarity), and microbial diversity covered.

	Aseel	Broiler	Ghagus	Nicobari
**Total Sequences**	2,135,894	493,732	484,707	502,563
**Average Sequences/Sample**	266,987	61,717	60,588	62,820
**Average No. OTUs (*n* > 2)**	3461	857	1911	1767
**Average Microbial Diversity (% Good’s Coverage)**	99.1	98.6	97.2	97.7

**Table 2 microorganisms-09-00391-t002:** Statistical analysis of alpha diversity measures for comparing different breeds or line. *p*-values were adjusted for false discovery rate (FDR).

Metrics	Mean ± SD Values ^#^				Kruskal–Wallis Test (*p*-Values)	Dunn Post-Hoc Test—FDR Adjusted *p*-Values					
	B	A	G	N		B vs. A	B vs. G	B vs. N	A vs. G	A vs. N	G vs. N
**Richness**											
**Observed richness**	513.9 ± 147	902 ± 99	984 ± 243	857 ± 131	0.001	<0.001	<0.001	0.002	0.663	0.688	0.645
**Chao1**	717 ± 119	1653 ± 223	1601 ± 371	1450 ± 246	<0.001	<0.001	<0.001	<0.001	0.195	0.606	0.721
**ACE**	755 ± 103	1691 ± 255	1704 ± 392	1557 ± 274	<0.001	<0.001	<0.001	<0.001	0.492	0.530	0.959
**Diversity**											
**Simpson**	0.938 ± 0.027	0.974 ± 0.015	0.973 ± 0.019	0.976 ± 0.010	0.011	0.018	0.020	0.014	1	1	1
**Shannon**	4.01 ± 0.54	4.91 ± 0.32	4.95 ± 0.41	4.83 ± 0.34	0.007	0.018	0.015	0.010	0.967	0.865	0.959
**Fisher**	116 ± 43	240 ± 35	274 ± 99	225 ± 46	0.001	<0.001	<0.001	0.002	0.492	0.530	0.645

^#^ Sequences were rarefied to the minimum library size (at 12,151 sequences per sample); A, Aseel, B, Broiler, G, Ghagus, N, Nicobari.

## Data Availability

The datasets presented in this study have been deposited in the Sequence Read Archive of the NCBI (accession numbers: PRJNA641245 and PRJNA641779).

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
