# Peer review of "Gut Microbial Composition Differs Extensively among Indian Native Chicken Breeds Originated in Different Geographical Locations and a Commercial Broiler Line, but Breed-Specific, as Well as Across-Breed Core Microbiomes, Are Found"

_microorganisms, 2021, doi:10.3390/microorganisms9020391_

Round 1

Reviewer 1 Report

Overall, the manuscript does not have any significance that would expand the knowledge in related fields. All results are easily expected and no further meaning/application can be found.

Reviewer 2 Report

This study describes a comparison of the gut microbiome of three chicken breeds indigenous to India, and one broiler chicken line. While many papers have been published on the chicken microbiota, indigenous breeds have been understudied. This study therefore presents an interesting description of some understudied breeds. The analysis methods used are appropriate and the conclusions drawn from the results are justified. My main criticisms are a lack of methodological detail regarding animal experiments, and the results and discussion sections being overly long and thereby obscuring the most important/novel results (please see specific comments below). There are several small errors in English language usage throughout, but these do not affect the ability of the reader to understand the meaning of the text.

Specific comments:

Introduction:

Line 35: It seems a little much to claim that chickens are “The cornerstone”, perhaps “a cornerstone” would be more appropriate.

Line 51-54: Please provide references for these statements, preferably with papers related specifically to chickens.

Line 67: Please provide a reference for this statement.

Line 83: “and is believed to have high disease resistance.” – Please provide a reference.

Materials and methods:

Line 103: At what age were the birds collected from their breeding tracts? Early life exposure to different environmental and nutritional conditions can greatly influence the microbiota. If the birds were reared for any length of time in these environments before being moved to the new facilities then the differences in environment in early life are likely to have impacted their microbiota composition.

Line 108: Please provide more details on husbandry conditions:

  • Did chickens receive any vaccinations?
  • How many chickens per pen?
  • How large were pens?
  • The full composition of the diet.
  • Were measures put in place to prevent faecal contamination between pens, which may have led lines to be exposed to microbiota from other lines?

Line 114: I understand that commercial broilers and indigenous chickens grow at very different rates and therefore reach slaughter age at different times. However, I am concerned that by collecting samples from commercial birds at 6 weeks and indigenous birds at 35 weeks the differences observed between them may also be a factor of age, not just line.

Line 115: How large were the initial groups? Were all of the birds for one line selected from one pen, or multiple pens? Were there any differences in body weight between the groups?

Line 117: Why were the gut contents mixed, rather than storing them separately: ie. duodenum, ileum, caeca etc? By mixing them it is not possible to see which sections of the gut are most affected by line differences. As the caecum contains by far the highest concentration of bacteria it would also be expected that the DNA from this section would drown out the relatively smaller amount of DNA from the other sections.

Line 120: Were negative controls or mock community controls included, to control for PCR bias and contamination?

Line 131: “ with suitable adapters.”: Please specify.

Lien 142: This reference is not for the mothur software. Please cite the references specified by the mothur creators on their website (https://aem.asm.org/content/75/23/7537.short and https://aem.asm.org/content/79/17/5112) and the date on which the Miseq SOP was accessed.

Line 148: Please provide reference for SILVA.

Line 151: “2 differences”: 2bp differences?

Line 155; “unrelated taxon”: Does this refer to taxa that were not bacteria or archaea?

Line 159: Please provide reference for Greengenes.

Results:

The results section is very long, due to the vast amount of detail included. This means that the more interesting findings of the study are not highlighted as they are hidden between less informative results. Some recommendations:

  • Correlation network analysis could be represented as figures only, with limited amounts of text.
  • The amount of taxonomies analysed could be decreased. For example, in many sections there are analyses of OTUs, phyla, family, order and genus. I would select one or two of these which the authors think are most relevant for that particular analysis. The other taxonomies can be fully described in supplementary figures and tables.
  • Line 252-295: This information would be easier for the reader to quickly understand, or to scan for the information they were particularly interested in, if it were presented as a figure or table.
  • Line 370-392: This paragraph could be condensed as much of the information it contains can be seen in Figure 3.

Table S1: The inclusion of standard deviations alongside means would allow the reader to gain an understanding of the variability within lines.

Line 297: “Aseel, Broiler, Ghagus, and Nicobari had 263,172,249 and 230 phylotype-OTUs, respectively.” I am unsure what these numbers refer to.

Line 496: Why were these parameters chosen to decide what is and is not a member of the core microbiome?

Line 570: Why is the family level analysis included in a different section from the genus and order analysis (line 616)?

Figure S8: Due to the complexity of this figure it is not possible to distinguish which bacteria are connected to others The Legend also states that this figure shows orders, but the labels within the figure are genera.

Discussion:

As with the results section, this section is very long and it is difficult for the reader to pick out the most pertinent or novel information (ie. similarities between indigenous breeds in comparison to the broiler line, comparisons to previous studies of indigenous birds, the core microbiome of the indigenous birds etc.). I would therefore advise trimming the discussion to include only discussions of those results that the authors feel would be of most interest to the reader. Methodological information such as in lines 717-735 can also be removed.

Line 637- 649: Much of this information has already been presented in the introduction. I would recommend removing this text.

Line 774: Throughout the text, please include references wherever claims of beneficial effects of particular microbes are listed.

Author Response

Kindly see the attached file

Reviewer 3 Report

In presented manuscript, the composition gut microbiome of 3 indigenous Indian chicken breeds (Nicobari, Ghagus, and Aseel )  and commercial broiler hybrid VenCobb 400 were analyzed.

As I reviewed few articles ( microbiome analyses) with very poor bioinformatics and statistic treatments in last time, I must admire precise analysis in presented manuscript. However it looks like that authors flood down into the statistics without bringing the clear, simple, and easy to understand message through whole manuscript. Instead of simple clear results the same data are processed by various similar statistics and differences are lengthy described in results and differences among used statistics (e.g. Unifrac) are disputed in discussion Moreover many information are doubled – they are in text an also in figures.. Tables from supplement should be incorporated to main manuscript and then text can be simplified, shortened and pointed out

From methodical point of view, I found 2 main faults inside this manuscript

First is the use of chickens with different age. Microbiome is age related and difference between broiler (6 weeks) and indigenous breeds (35 weeks) is enormous. I am sure that his is a source of big portion of differences among them. Moreover, authors only partially discuss age as source of variability with statements about microbiome stabilization at 3 weeks. But this seems to be cherry picking as there are many studies about age/microbiome relation in birds.

Second problem is related to increased richness in Aseel gut microbiome. This increase may be related to high number sequences/ sample. No mater of clustering method or filtration higher sequence counts consistently brings increased richness in Illumina amplicon sequencing, thus richness may be artificially increased as Aseel samples had 5 time more sequences. I suggest that authors may try to decrease the number of input sequences in order to obtain comparable data.

Authors showed the existence of breed specific complex microbial networks in all groups. However microbial network on figure S8 is hard to view and understand. While I doubt about the need of 4 NMDS analyses in figure 3 (one is sufficient); I recommend division of figure S8 into 4 smaller figures each for particular breed. Figure 7 may be also divided, but it is readable in current form. Please consider also stacked bar charts for all samples at reasonable phylogenetic levels (phylum, class) instead or in addition to Krona

Based on aforementioned facts major revision needs to be done in manuscript

For publication of manuscript:

Authors must explain why they used subjects of different age and thoroughly discuss it or remove the broiler from analysis.

Authors must be more concise in text, mainly in results

Authors should look over sequencing depth in relation to increased richness

Round 2

Reviewer 1 Report

Thank you for your effort to improve the manuscript. One thing I concern is that the age of the experimental animals (chicken/broiler) seems to be different. It is generally recognized that the microbiome profiles of the animals in different ages are quite distinct. Isn't it?

Thanks,

Reviewer 3 Report

Authors made changes according suggestions.

They also explained reason why  different age of chicken was used. Despite this is the main flaw of this study, it is at least correctly discussed. Thus manuscript meets criteria for publication.